## RESEARCH ARTICLE

# Ultraviolet B radiation impairs coral reef fish development

**Adam T. Downie\*, Coen Hird, Rebecca L. Cramp, Fabio Cortesi and Craig E. Franklin**

## ABSTRACT

Loss of structural habitat complexity associated with habitat degradation in marine systems may expose early life stages of fishes to harsh environmental conditions. Specifically, loss of coral cover means less suitable refuge is available for some reef fish species to lay their eggs, exposing them to pervasive stressors such as ultraviolet radiation (UVR). Here, using laboratory experiments, we exposed embryos of the clownfish *Amphiprion ocellaris* for 2 h daily to two UVR levels reflective of their depth at settlement; high UVR (280 µW m$^{-2}$), reflective of shallow depths, and low UVR (80 µW m$^{-2}$), reflective of deeper depths over their embryonic period, and then measured changes in mass, yolk sac volume, DNA damage, and survival. Despite being exposed to ecologically relevant levels of UV radiation, there was 100% mortality before hatching and inflated yolk sacs in both high and low UVR-treated animals. Exposure to UVR also resulted in DNA damage, albeit only in high UVR treatments. It is evident from our results that the protection that the reef can offer from UVR is critical for the survival of clownfish. Our results also underscore the need for future work to consider this often-neglected stressor and the role of adequate refuge for the healthy development of early-life stages of reef organisms.

KEY WORDS: Habitat degradation, Early life history, Embryos, Eggs, DNA damage, Clownfish, Environmental stress

## INTRODUCTION

Habitat loss is occurring on a global scale, primarily driven by increased human activity (Hodapp et al., 2023; Powers and Jetz, 2019). A consequence of habitat loss is decreased shelter and refuge availability that would otherwise protect fauna from environmental conditions and predators. While mass deforestation of temperate and tropical rainforests is typically associated with habitat loss, oceans are also currently facing large-scale habitat destruction (McCauley et al., 2015). Specifically, ocean warming and the increased frequency and intensity of storms associated with anthropogenic climate change are causing significant decreases in live coral cover on reefs (Hughes et al., 2017). Since hard corals provide structure and refuge for many reef-associated animals, including invertebrates and fishes, their loss poses significant challenges to survival. A critical yet often over-looked environmental factor that is likely to have a substantial impact on fishes on a barren reef is ultraviolet radiation (UVR; Hird et al., 2024).

UVR from sunlight (100-400 nm) is a pervasive diurnal environmental driver acting on terrestrial and aquatic ecosystems, with the exception of subterranean habitats and marine depths exceeding 50 m (Alves and Agustí, 2020; Lee et al., 2013). UVR intensity is generally highest at low latitudes where most coral reefs occur and can penetrate up to 30 m in clear water (Banaszak and Lesser, 2009; Downie et al., 2024). Biologically, UVR is essential for bone metabolism for many vertebrate species and for sterilising viruses (Abshire, 1987; Adams et al., 1982). Conversely, UVR is both cytotoxic (increases production of oxygen radicals, which causes oxidative damage to cells and metabolites) and genotoxic (directly damages DNA) (Downie et al., 2023b). While UVR typically targets cellular processes and molecules, the impacts at the cellular level cascade to impact the whole animal, resulting in several sub-lethal and lethal consequences on animal fitness, such as delayed developmental stages (Lundsgaard et al., 2021, 2024), decreased locomotion (Ghanizadeh Kazerouni et al., 2016), and mortality (Holmquist et al., 2014). Tropical fishes have evolved many strategies to cope with high UVR intensities, such as biological sunscreens in the skin and eyes (mycosporine-like amino acids, MAAs; Mason et al., 1998), antioxidant defences (Banaszak and Lesser, 2009), photo-repair mechanisms (Dethlefsen et al., 2001; Dong et al., 2007), and behaviourally avoiding UVR by seeking refuge under tabular corals (Kerry and Bellwood, 2015a,b, 2017). These biochemical and behavioural strategies are usually present in adult fishes, but early-life stages may lack many of these defences.

Fish embryos and larvae are typically the most vulnerable life stages to most environmental stressors (Downie et al., 2020). The impacts of increased UVR on fish embryos are less well-documented compared with other environmental factors that are rapidly changing, such as elevated temperature (Alves and Agustí, 2020). However, several studies focusing on temperate fish species, such as red sea bream (*Pagrus major*), Atlantic cod (*Gadus morhua*), and black sea bream (*Acanthopagrus schlegeli*) have revealed that embryonic exposure to UVR can have both sub-lethal and lethal consequences (Fukunishi et al., 2010; Kouwenberg et al., 1999; Lesser et al., 2001). For example, increased UVR exposure may delay or reduce hatching success (Alloy et al., 2016; Charron et al., 2007; Fukunishi et al., 2010; Icoglu Aksakal and Ciltas, 2018; Mahmoud et al., 2009; Vásquez et al., 2016), increase oxidative damage to cells (Charron et al., 2007; Lesser et al., 2001; Mekkawy et al., 2010), increase malformations (Icoglu Aksakal and Ciltas, 2018; Mahmoud et al., 2009; Sayed and Mitani, 2016; Vásquez et al., 2016), increase DNA damage (Lesser et al., 2001; Mekkawy et al., 2010), or increase mortality (Holmquist et al., 2014; Kouwenberg et al., 1999; Lesser et al., 2001; Mahmoud et al., 2009). Whether the embryos of tropical reef fishes show similar responses to UVR exposure remains poorly understood. Considering their habitat inherently shows very high UVR levels, we might expect greater resilience in coral reef fishes.

School of the Environment, The University of Queensland, St Lucia, Brisbane, 4072 Australia.

*Author for correspondence (adam.downie@qut.edu.au)

A.T.D., 0000-0001-7981-0724; C.H., 0000-0001-9812-4818; R.L.C., 0000-0001-9798-2271; F.C., 0000-0002-7518-6159; C.E.F., 0000-0003-1315-3797

In some reef environments, UV intensities may be as high as 2 W m$^2$ and 40 W m$^2$ for UVB (280-315 nm) and UVA (315-400 nm), respectively (Torregiani and Lesser, 2007). In contrast, UVB levels in many temperate aquatic ecosystems range from 0.1 W m$^2$ (Häkkinen et al., 2002) to 0.7 W m$^2$ (Markkula et al., 2009). UVR is expected to be highest in shallow waters and progressively attenuated with increasing depth (Dunne and Brown, 1996). While many tropical species broadcast thousands or millions of pelagic eggs in a spawning event, some tropical species, such as clownfishes (Amphiprioninae, Pomacentridae), lay fewer (hundreds), well-developed eggs among anemones and coral to protect them from environmental conditions, including UVR (Pacaro et al., 2023). While pelagic eggs are exposed to UVR generally at the water's surface, some species have been found to biochemically initiate negative buoyancy when UVR exposure crosses a threshold (Dethlefsen et al., 2001; Pasparakis et al., 2019). In contrast, benthic eggs are typically adhered to the substrate and are unable to move from stressors, relying solely on structures like coral and anemones for shelter. However, if this structure is unavailable due to habitat loss, adults may lay eggs in sub-optimal habitat patches, and adhered eggs may receive direct exposure to UVR. This may disproportionally affect individuals inhabiting shallow depths of their habitat range compared to those living at deeper depths. Embryos in UVR-exposed conditions may accumulate more DNA damage, e.g. cyclobutane pyrimidine dimers (CPD), a primary form of UV-associated DNA damage in some organisms (Hird et al., 2023; Lesser et al., 2001) or deploy mechanisms to mitigate DNA damage; both scenarios may place stress on available energy reserves. For example, under high temperatures, herring (Clupea harengus) yolk sac volume is decreased due to the increased energy demand required to develop under warmer conditions (Toomey et al., 2023). UVR effects on energy reserves in tropical fishes are largely unknown but may contribute to decreased hatching success.

Here, we used the false percula clownfish (Amphiprion ocellaris) as an example coral reef organism that lays benthic eggs, to determine the impacts of UVR exposure on the development of coral reef fish embryos. A. ocellaris occupies a depth range of 1 to 15 m (Madduppa et al., 2014), and we used the upper- (2 m; UVB: 280 µW m$^{-2}$; high UVR treatment) and mid-depths (6 m; UVB: 80 µW m$^{-2}$; low UVR treatment) of this range as our UVR treatment groups (Braun et al., 2016), along with an experimental control exposed to no UVR. Embryos were exposed to UVR for 2 h daily, representing the peak irradiance period on reefs. Exposure was conducted from laying through their entire embryonic period (typically 8 days) to evaluate changes in mass, yolk volume, DNA damage, and survival. We hypothesised that embryos exposed to high UVR will have significantly reduced survival, decreased yolk volume, decreased mass, and increased DNA damage than embryos exposed to low or control conditions.

## RESULTS

The most significant impact of UVR was on embryo yolk sac volume (Fig. 1B) and survival (Fig. 2). At 2 days post-fertilisation (dpf), average embryo yolk sac volume was 29% lower under control conditions than under high UVR (HPDI$_{lower}$=−0.36, HPDI$_{upper}$=−0.034), and yolk sac volume was 48% lower under control conditions than low UVR treatments (HPDI$_{lower}$=0.48, HPDI$_{upper}$=−0.15; Fig. 2B). Under control conditions, there is a decreasing trend, albeit non-significant (10% decrease; HPDI$_{lower}$=−0.0048, HPDI$_{upper}$=0.25; Fig. 2B), in yolk sac volume between 2 and 5 dpf as the embryo absorbs the yolk to develop tissue

and organ systems (Salis et al., 2021). However, at 5 dpf yolk sac volume for both low and high UVR treatments were significantly higher than control embryos at 5 dpf (5 dpf control versus 5 dpf low, 61% higher; HPDI$_{lower}$=−0.48, HPDI$_{upper}$=−0.15; 5 dpf control versus 5 dpf high, 38% higher; HPDI$_{lower}$=−0.36, HPDI$_{upper}$=−0.034; Fig. 2B).

Generally, embryo mass did not differ between treatments and age (Fig. 1A). The only observable difference in mass was at 5 dpf when the embryo mass under control treatment was 12% higher than those under the high UVR treatment; however, this was not statistically different (HPDI$_{lower}$=−0.047, HPDI$_{upper}$=0.21).

Interestingly, there was no measurable DNA damage in low UVR-exposed embryos. Contrary to our hypothesis, there was a slight reduction in the extent of DNA damage as embryos aged although this relationship was not significant (HPDI$_{lower}$=−0.29, HPDI$_{upper}$=0.98; Fig. 1C).

Aligning with our hypothesis, the survival of embryos was significantly impacted by UVR exposure (Fig. 2). Embryos from all three replicates of the control treatment survived until hatching at 9 dpf and experienced no daily mortalities (Fig. 2). Cumulative survival was variable among treatments and replicates, but both UVR treatments resulted in 100% mortality before hatch (Fig. 2).

## DISCUSSION

Exposure to UVR had significant consequences for the embryonic stage of the false percula clownfish, causing complete mortality independent of exposure levels. This was unexpected, as we assumed UVR would cause damage but not the complete loss of a clutch, especially at lower exposure. Contrary to our hypotheses, we found that embryo exposure to high and low UVR resulted in increased yolk sac volume at 2 dpf and 5 dpf with no significant changes in mass or DNA damage observed. Therefore, early UVR exposure, especially under habitat loss conditions, may have significant lethal consequences on the survival and development of marine life, such as anemonefishes, whose embryos develop on benthic habitats that rely on shelter from UVR.

UVR had an impact on the development of clownfish embryos, particularly yolk volume. Several studies have shown that UVR exposure causes a range of significant morphological defects and malformations in embryos, including notochord abnormalities, fin blistering, eye deformities, and dwarfism (Dong et al., 2007; Mahmoud et al., 2009; Mekkawy et al., 2010; Sayed and Mitani, 2016). While few studies have considered the effects of UVR on embryo yolk reserves, yolk sac deformities, such as inflated yolk sac oedemas and kyphosis, have been found in African sharptooth catfish (Clarias gariepinus) and Japanese medaka (Oryzias latipes) embryos exposed to elevated UVR (Mahmoud et al., 2009; Sayed and Mitani, 2016). Additionally, yolk sac diameter was also found to be significantly larger in both Pacific mackerel (Scomber japonicus) and northern anchovy (Engraulis mordax) exposed to UV compared to controls (Hunter et al., 1979). Disruptions in ion balance, bacterial infections, and physical trauma from environmental stressors like temperature, salinity, and UVR typically cause yolk-sac oedemas/enlargements in fish embryos (Chandra et al., 2024). In the current study, it is possible that UVR may have denatured proteins and enzymes that maintain osmotic balance in the yolk sac, causing inflation; however, the yolk contents would have to be measured to confirm. Alternatively, UV-induced DNA damage may have caused direct transcriptional disruptions impacting normal development. For example, UVR exposure inhibited the expression of genes related to cell growth and differentiation in juvenile gilthead seabream (Sparus aurata) (Alves and Agustí, 2022). Disruptions

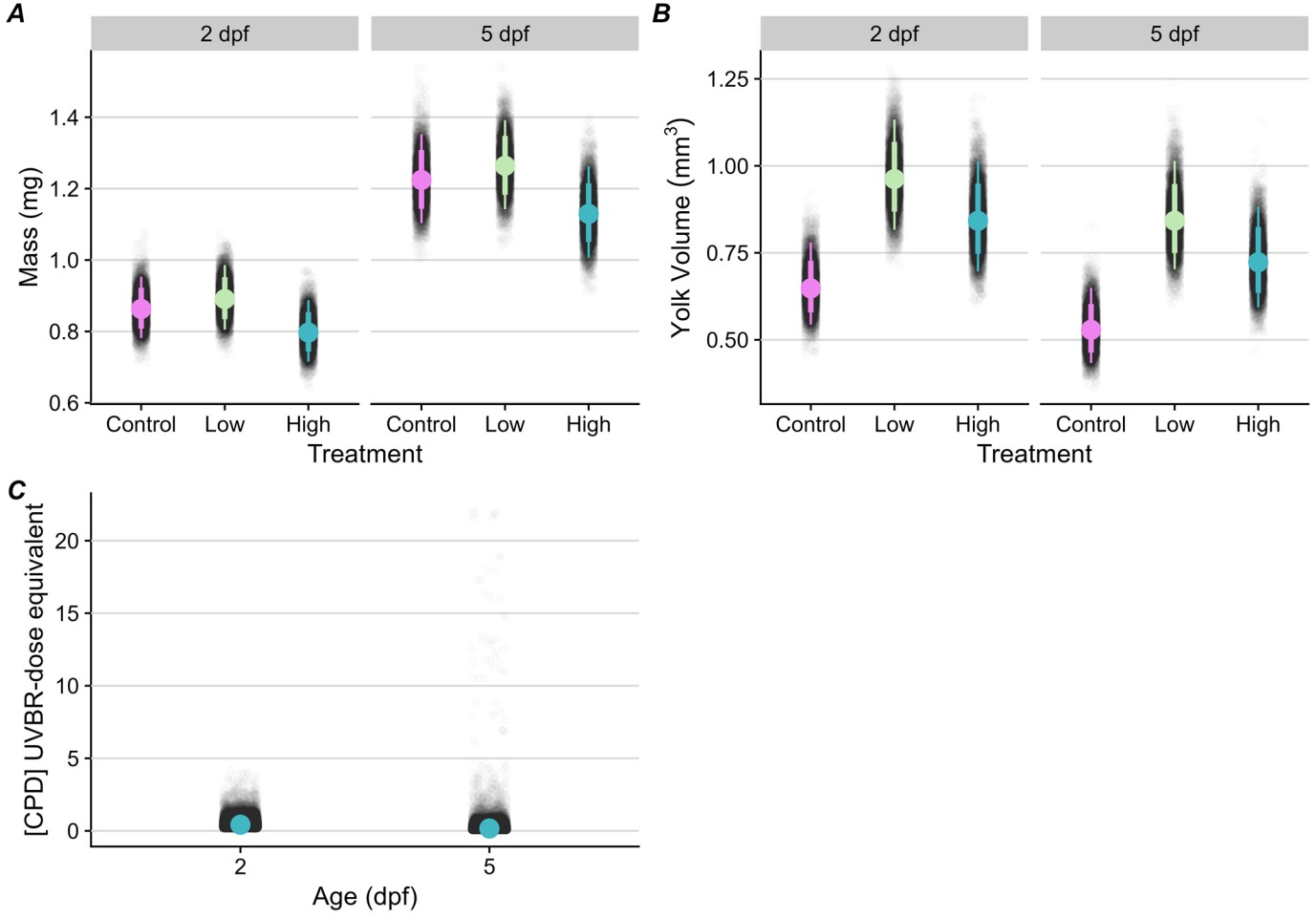

**Fig. 1. The effect of UVR (UVB: 280-315 nm) across developmental age (dpf) of clownfish (*A. ocellaris*) embryos.** Embryos were exposed to either control (no UVR; pink colour bar) or UVR levels representing the within-depth range of *A. ocellaris*: 6 m (low UVR treatment; 80 µW m²; green colour bar) or 2 m (high UVR treatment; 280 µW m²; blue colour bar) (Braun et al., 2016). UVR exposure occurred for 2 h daily from when eggs were laid (0 dpf) until hatched at 8-9 dpf. On 2 and 5 dpf, individual embryos were collected for (A) mass (mg; *n*=15 per age per treatment group), and (B) yolk sac volume (mm³; *n*=15 per age per treatment group). (C) DNA damage (concentration of CPD) only was measurable in embryos exposed to high UVR treatment only (*n*=13 measured in 2 dpf embryos, and *n*=8 measured in 5 dpf embryos). The experiment was replicated three times per treatment using two different parent pairs. In all three panels, predicted Bayesian posterior distributions are shown with symbols and error bars representing the median and 0.95 and 0.8 quartile-based credible intervals, respectively.

of UVR on the transcriptome-level impacting embryo development have been found in other organisms as well, such as decapods (dos Santos et al., 2020). It is possible that UVR slowed the progression of developmental stages, as has been previously found in amphibians (Lundsgaard et al., 2021, 2024), which would mean less yolk was used. However, morphological developmental milestones characterising embryos at 5 dpf were consistent across control, UVR and high UVR treatments, such as black pigmentation of the eyes and mouth opening (Salis et al., 2021). As reef fish early life development is highly dynamic at the molecular level (Downie et al., 2023a; Huerlimann et al., 2024; Roux et al., 2023), a transcriptomic analysis could reveal other changes that may have contributed to yolk-usage (e.g. upregulation of photo repair) under high UVR, besides morphological changes. Interestingly, low UVR had a greater effect on yolk volume than high UVR (low and high UVR-treated embryos had 61% and 38% greater yolk volume than controls, respectively). This may be attributed to the initial biochemical response of embryos to high versus low UVR. Higher UVR dosages may have elicited a greater initial biochemical response to UVR and therefore invested more energy in antioxidant

production, DNA damage and photorepair. In contrast, low UVR exposure may elicit less energy-intensive responses; however, these responses accumulate over the entire embryonic cycle, resulting in an overall high mortality rate. Additionally, it is possible that yolk sac absorption was reduced if the embryo was dying from UVR exposure for both low and high UVR exposure intensities.

Contrary to our hypotheses, only fish exposed to high UVR had any measurable DNA damage, with no individuals sampled from control or low UVR treatments having any measurable DNA damage. However, CPD concentration was highly variable, which may have contributed to mortality (see below). The lack of DNA damage in the low UVR treatment and the highly variable amounts of damage with high UVR exposure suggests that clownfish embryos may have employed DNA photoprotection, such as increased gene expression of photo-repair mechanisms (e.g. photolyases), antioxidants (e.g. superoxide dismutase), or sunscreen/pigments in the skin and egg yolk, similar to those exhibited in tropical *Mahi mahi* (Pasparakis et al., 2019). Additionally, the decreasing trend of DNA damage with age under high UVR conditions suggests that DNA photoprotective

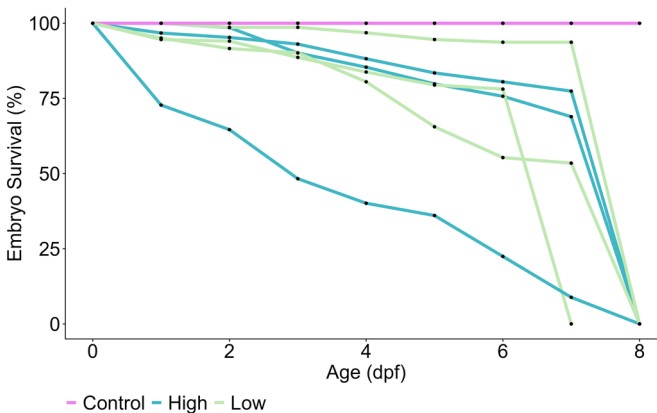

**Fig. 2. The impact of UVR (UVB 280-315 nm) on embryo cumulative survival (%) across developmental age (dpf) of clownfish (*A. ocellaris*).** Embryos were exposed to either control (no UVR; pink line) or UVR levels representing the within-depth range of *A. ocellaris*: 6 m (low UVR treatment; 80 µW m$^2$; green line) or 2 m (high UVR treatment; 280 µW m$^2$; blue line). UVR exposure occurred for 2 h daily from when eggs were laid (0 dpf) until hatching at 8-9 dpf. The experiment was replicated three times per treatment using two different parent pairs, represented by three lines per treatment.

mechanisms improve or increase as embryos age. DNA damage and oxidative stress have been measured in response to UVR in temperate and sub-tropical species (Applegate and Ley, 1988; Cha et al., 2012; Mahmoud et al., 2009; Mekkawy et al., 2010). However, some temperate species, like flatfish dab (*Limanda limanda*) and plaice (*Pleuronectes platessa*), demonstrated considerable photo repair when the buoyancy of the yolk sac decreased due to UVR damage (Dethlefsen et al., 2001). Our results suggested that at an early stage, tropical reef species may have developed some biochemical defences to combat UVR-induced DNA damage better than predicted. However, as we observed in this study, such biochemical defences may not always reduce UVR-associated mortality.

In support of our hypothesis, UVR had significant impacts on *A. ocellaris* embryo survival, which is also commonly found in embryos of other fish species exposed to UVR (Holmquist et al., 2014; Kouwenberg et al., 1999; Lesser et al., 2001; Mahmoud et al., 2009). Under control scenarios, embryos experienced no mortality. When provided adequate aeration, it is common for all clownfish embryos to survive and hatch (A.T.D., personal observation). Similar experiments exposing amphibians to UVR have also found that under control conditions, 90-100% of individuals survived (Lundsgaard et al., 2020, 2024). In stark contrast, under both high and low UVR, clownfish embryos experienced 100% mortality. Typically, mortality rates of fish embryos exposed to UVR are highly variable and are also dependent on UVR dosage and duration of experiments. For example, embryos of woundfin (*Plagopterus argentissimus*) and yellow perch (*Perca flavescens*) experienced 100 and 98% mortality, respectively, when exposed to UVR (Holmquist et al., 2014). On the contrary, mortality of anchovy embryos (*Engraulis ringens*) ranged from 24% to 53% (Vásquez et al., 2016), depending on embryonic stage and UVR dosage (UVA only versus UVB and UVA). Atlantic cod (*G. morhua*) embryos experienced 80% survival when exposed to UVR intensity of 6.38 Wm$^2$ (total dosage of 472 kJ) for 30 min, and 27% survival when exposed to UVR intensity of 4.04 Wm$^2$ (total dosage of 116.4 kJ) for 2.5 h (Kouwenberg et al., 1999). Tropical and subtropical species also have been found to have high embryonic mortalities when exposed to UVR, such as up to 75% for zebrafish

(*Danio rerio*; Dong et al., 2007) and up to 90% for the common sardine (*Strangomera bentincki*; Vásquez et al., 2016). Given the variable response of fish embryo survival to UVR, species-specific responses to UVR at the cellular level are likely at play (e.g. photo-protection, antioxidant responses, use of alternate metabolic pathways to supplement energy in response to stress). Interestingly, both low and high UVR treatments resulted in 100% mortality in clownfish embryos, with most mortalities occurring before hatching. Similar to what is causing the differences in yolk volume between high and low UVR treatments, it is possible that the higher UVR treatment elicited a more immediate biochemical response, which may have increased energy use early on, resulting in less energy for survival as the embryos approached hatching. In contrast, the low UVR treatment is likely to have caused a more gradual accumulation of damage that surpassed a tolerance threshold at hatching. It is also likely that hatching enzymes in both treatments were damaged or depleted, which would cause a systematic loss of the clutch (i.e. total mortality).

While we cannot determine the exact factors that influenced mortality, inflated yolk sacs possibly contributed. When ready to hatch, clownfish embryos use hatching enzymes and mechanical motion (from their caudal muscle and tail motion from parents) to break from their egg casing (Yamanaka et al., 2024). An inflated yolk sac may have made it physically challenging for the embryo to break free from the egg casing, trapping the embryos inside the eggs. Additionally, embryos that received high-UVR exposure showed a trend for decreased mass (albeit not statistically significant), which may have resulted in less musculature to break free from the egg mechanically. Alternatively, variable individual responses to DNA damage may have also contributed to mortality. Since we only measured DNA damage up to a few days before hatching (5 dpf), cumulative DNA damage at 8 dpf may have significantly contributed to mortality. Any intrinsic UVR defences or photo-repair may have become energetically expensive by the time of hatch, (i.e. DNA damage at 8 dpf may have been more significant than at 5 dpf, despite the decreasing trend in damage from 2 dpf to 5 dpf). Given that UVR is both genotoxic and cytotoxic, other cellular and molecular processes may have been impaired, which contributed to the elevated mortality rate in exposed embryos. For example, UVR may have denatured the hatching enzymes that help dissolve the egg casing, or oxidative stress (e.g. ROS build-up) causes additional cellular stress, diverting energy from development and hatching.

Our results contribute to the growing body of literature that suggests that clownfishes are highly sensitive to environmental change across life history stages, and reproductive output may be impacted by environmental stressors. Indeed, scientists have only just started to investigate the direct impact of coral cover, anemone availability, and habitat quality on clownfish reproduction and carry-on effects for future generations (e.g. Salles et al., 2020; Cortese et al., 2021). Macro-ecological studies have found that clownfish populations are susceptible to acute, small-scale changes in habitat structure and quality, such as decreases in anemone abundance associated with ocean warming and irradiance (Hobbs et al., 2013; Salles et al., 2020). This loss of shelter would presumably expose early life stages to adverse environmental conditions. Bleaching is a significant threat to coral reefs and contributes significantly to increased habitat degradation of the world's coral reefs (Hughes et al., 2017). Recent evidence suggests that under anemone bleaching conditions, fecundity is decreased by up to 73% (Beldade et al., 2017). Additionally, adult clownfishes spend less time in bleached anemones (Cortese et al., 2021), which

would have ramifications on the parental investment in embryo development. A pervasive stressor such as UVR is likely to compound the effects of bleaching and habitat degradation, especially if eggs are directly exposed to the radiation. Future work evaluating the reproductive output of marine life under various degrees of habitat degradation and environmental stressors (including UVR) will provide valuable insights into the significance of shelter for preserving populations under stress. This study is among the first to investigate UVR effects on coral reef fish embryo development and contributes to the understanding of the complex interactions of multiple environmental stressors associated with continued coral reef habitat loss and their potential ramifications on fish population dynamics. Our results contribute to an increasing body of literature demonstrating that UVR is an important, yet often overlooked, environmental stressor impacting fish life history (Hird et al., 2024). We show that even low levels of UVR impact fish embryonic development, meaning individuals living near the deeper parts of their depth range may still be at risk. This emphasises the importance of the physical structure of reefs and the refuges provided by the complex morphologies and geometries of corals and other organisms like anemones. However, habitat loss due to increased storms and thermal stress puts such refuges at risk, limiting their availability and increasing competition for these limited spaces (Kerry and Bellwood, 2015a,b, 2017). UVR has been found to interact with multiple environmental stressors, including temperature, in marine, freshwater and terrestrial systems (Downie et al., 2024; Lundsgaard et al., 2023). Therefore, we call for future work to routinely include UVR as a daytime stressor on reef ecosystems when planning experiments to explore the effects of environmental change. Studies assessing the impact of UVR exposure should also be expanded to incorporate its effects across life history and generations and from the individual to the population levels.

## MATERIALS AND METHODS

### Ethics statement
Clownfish breeding pairs are kept under Animal Ethics 2022/AE000638, and embryo exposure to UVR experimental protocol was approved under Animal Ethics 2022/AE000454. Both ethics were reviewed and approved by the University of Queensland's Animal Ethics Committee.

### Animal husbandry and embryo collection
Captive breeding pairs of *A. ocellaris* were maintained in recirculating aquaria (volume=95 litres) at The Institute for Molecular Biosciences (The University of Queensland). A single terracotta pot (27 cm diameter) was placed in the tank for refuge, environmental enrichment, and to provide a place for females to adhere eggs. Pairs were fed twice daily using commercial fish pellets (Ocean Nutrition Formula One Marine Pellet Small; https://oceannutrition.com/). Water temperature was maintained at 28°C during summer, reflecting natural breeding conditions (Roux et al., 2021), using a series of heaters and chillers. The adults were exposed to a natural photoperiod of 12 h light and 12 h dark provided by ceiling-mounted fluorescent (i.e. non-UVR emitting) bulbs. Females typically lay eggs fortnightly in the late afternoon to early evening (15:00-18:00). Eggs were adhered under the lip of the pot, and the male would subsequently fertilise them. While there were several breeding pairs in the colony, we took eggs from two breeding pairs for consistency, which limited potential genetic variability and allowed for uninterrupted experimental flow (i.e. reliance on only one pair would result in waiting approximately 2 weeks for a new clutch).

Every morning, terracotta pots were visually checked for eggs. Newly fertilized eggs are a light pink with a bright white vegetal pole, while unfertilized eggs appear beige in colour (Salis et al., 2021). The male clownfish recognises unfertilised eggs and removes them swiftly during mating. Egg masses used in the experiments were typically all pink with white caps, indicating that all eggs were fertilized. Upon discovering a new clutch, the pot was removed from the tank, and the eggs were removed from the pot by fragmenting a shard of the pot with the eggs on it. Effort was made that fragmenting the pot did not create a fissure through the clutch. Instead, the pot was broken around the clutch so that it was a small enough size to fit into the experimental tank, as the experimental tank was too narrow to accommodate the entire pot. The fragmentation of the pot did not result in embryo mortality. The isolated embryos were incubated in a separate flow-through experimental tank (volume=95 litres; water temperature maintained at 28°C; see 'UVR exposure procedure' for details) from adults and dosed with Methylene Blue to mitigate fungal infections (0.7 mg l$^{-1}$, Aquasonic). A photograph was taken 0 dpf to determine the number of eggs laid and fertilised.

### UVR exposure procedure
Embryos remained in the experimental tank for the duration of the experiment (0-8 dpf) and were exposed to the same water conditions and photoperiod as the adults. A single egg clutch of embryos was exposed to UVR at any one time in the experimental tank. The internal compartment of the exposure tank consisted of a small, perforated platform that held the shard with the eggs (and kept embryos approximately 20 cm off the bottom of the tank), an aerator that bubbled in between the perforations in the platform (which replicated the parents aerating the embryos), and an overhead UV lamp. The UV lamp (Exo Terra Sunray Fixture; Exo Terra) was positioned approximately 50 cm above the embryos. This distance was selected so target UVR dosages could be delivered to embryos and that the heat radiating from the bulb did not result in measurable increases in water temperatures. The two UVR treatments were low (UVB irradiance: 80 μW m$^{-2}$) and high (UVB irradiance: 280 μW m$^{-2}$), representing the shallow (high UVR; 2 m) and mid depths (low UVR; 6 m) of *A. ocellaris* depth range, based on measurements taken from the field on coral reefs (Braun et al., 2016).

Control embryos were placed in the experimental tank without UV lamps turned on (i.e. UVB: 0 μW m$^{-2}$) during the designated exposure times (see below). Because low levels of UV lighting were always present in the aquarium room (rearing tanks in the same room had UVR lights installed and emitted low levels of UVA and UVB), control tanks were covered with a sheet of UV-absorbing plexiglass placed on top of the tank. Additionally, the sides of the tank were covered with black plastic to prevent any potential UVR from entering from the sides of the tank. Hence, control embryos were exposed to the same photoperiod as the two treatments, experiencing a UV-deprived light habitat in the human visible spectrum produced from the fluorescent ceiling lights. A radiometer/photometer (IL1400BL, International Light Inc., Newburyport, USA) was used to calibrate UVR levels at the water's surface for all experimental groups. Aeration and Methylene Blue did not impact UVR penetration into the water, as UVR intensities were measured with and without these factors. Since the UV lamp remained in a fixed position across all experiments, sheets of thin transparent plastic (<1 mm thick) were placed on top of the experimental tank directly above the embryos to adjust UVR to desired irradiances: one sheet of plastic was used for high UVR-B treatments to obtain the target intensity of 280 μW m$^{-2}$, and five sheets of plastic were used for low UVR-B treatments to obtain the target intensity of 80 μW m$^{-2}$. Embryos were exposed to UVR for 2 h from 11:00-13:00, representing peak UVR exposure in nature (Braun et al., 2016). UVR exposure started on the day eggs were fertilized (0 dpf) until age at hatch (8-9 dpf). Each experimental group was replicated three times. A single clutch was used at a time for each experimental treatment replicate, spawned from one of two breeding pairs. Clutches were not divided as there was no guarantee that clutches would fragment evenly and directly fragmenting through the clutch may have caused mortalities.

Every morning (approximately 07:00), the terracotta fragments with developing embryos were gently removed from the experimental tank using a small dish containing water from the experimental tank and placed on a benchtop to evaluate mortalities under a benchtop dissecting microscope. Embryos were not exposed to air during this process, as transfer of the fragment with embryos into the small dish occurred underwater. Embryos are typically a light grey colour with the eyes having a

silvery sheen (Salis et al., 2021). Dead embryos were therefore easily identifiable because the body was white, and the eyes were no longer shiny. Dead embryos were counted and removed with forceps to measure cumulative mortality for each clutch each day. Removing the shard, counting/removing dead embryos and returning the shard to the experimental tank took <5 min, and did not result in mortality, as control treatments experienced no mortality (see Results section on mortalities). At 2 and 5 dpf, individual eggs were collected to determine embryo mass, yolk volume, and DNA damage (n=9 per trait per treatment replicate). These ages were selected as the developmental milestone of organogenesis is initiated at 2 dpf and completed at 5 dpf (Salis et al., 2021). At the end of the daily UVR exposure (13:00) at 2 and 5 dpf, terra cotta shards were immediately removed from the experimental tank and placed in a small tank with water from the experimental tank. Embryos were carefully removed from the egg casings using forceps to be evaluated for DNA damage and morphology. Embryos for DNA damage assays were immediately placed in dark tubes to prevent light-induced DNA photo-repair (Kelner, 1949; Rupert et al., 1958; Sancar, 2008) and stored at −80°C and embryos measured for mass and yolk volume were placed in phosphate-buffered seawater and stored in the fridge at 4°C.

## Morphology

Five embryos per day per experimental replicate were randomly selected to measure morphological traits (n=15 per treatment per day). Embryos were gently blotted dry and weighed on an electronic balance (mg). Individual embryos were photographed on top of a grid (1 mm gridlines) under a microscope. These images were transferred to ImageJ, where the grid sizes were used to calibrate the scale and measure the dimensions of the yolk sac. The length and height of the yolk sac were measured, and the yolk sac volume was calculated using the following equation:

$$V = \frac{\pi}{6} \times L \times H^2,$$

where V=yolk sac volume (mm³), L=length of the yolk sac (mm), and H=height of the yolk sac (mm) (Sariat et al., 2023). UVR has been found to reduce the progression of developmental stages in amphibians (Lundsgaard et al., 2021, 2024). Photographs at 2 and 5 dpf enabled us to determine if UVR reduced the progression of developmental stages based on developmental milestones described by Salis et al. (2021). Important milestones by 5 dpf include black pigmentation in the eyes, pectoral fin development, heart moves anterior, and mouth opening (Salis et al., 2021).

## DNA damage

Genomic DNA was extracted and purified from homogenates from single embryos using a Qiagen DNeasy Blood and Tissue Kit (Qiagen Inc., Hilden, Germany) and quantified using a Qubit dsDNA High-Sensitivity Assay Kit (Thermo Fisher Scientific Inc., Waltham, MA, USA). CPD concentration for each sample was determined using an anti-CPD ELISA assay following the primary antibody manufacturer's protocol (Mori et al., 1991) (NM-ND-D001, clone TDM-2, Cosmo Bio Co., Ltd.) and following Hird et al. (2023). Plates were read at 450 nm in a DTX880 multimode detector (Beckman Coulter, MN, USA) using the SoftMax® Pro program (Version 7.1.0, Molecular Devices LLC, CA, USA). CPD concentrations were calculated from a standard dose-response curve of UVC-irradiated calf thymus (NM-MA-R010, Cosmo Bio Co., Ltd.) present on each plate. CPD concentration is reported as units of UVCR-dose equivalents per 20 ng of DNA.

## Cumulative survival

Upon a clutch being laid (0 dpf) and the terracotta pot fragmented, a photograph was taken of the eggs on each shard. The eggs were counted, and at this time point, survival was 100%. Dead embryos were removed from the shards daily, and their number was subtracted from the previous day's total number. Cumulative survival for each experimental replicate (n=3 per UVR treatment per age) across the entire experimental

period (i.e. 0-8/9 dpf) was calculated as:

$$S_{a=} \frac{N_{(a-1)-M_a}}{C_{a=0}} \times 100,$$

where S=survival (%) at a given age a (dpf), N=the number of eggs alive from the previous day's age (a-1), M=the number of mortalities found on the current age, and C=the total number of fertilised embryos at age 0 dpf.

## Statistical analyses

All statistical analyses were performed in R (ver. 4.4.2) (R Core Team, 2021). Data were analysed using a Bayesian method using the 'brms' package (Burkner et al., 2021). All candidate models were run with four chains, 4000 (except for DNA damage, which was run at 18,000) iterations, a warm-up phase of 2000 iterations and non-informative priors. Model selection was based on leave-one-out cross-validation methods from the 'loo' package (Vehtari et al., 2022). 'Mass' was best described with a lognormal family, 'Yolk Volume' and 'DNA Damage' were best described with a Gamma (link='identity') family. UVR treatment and age were used as explanatory variables for 'Mass' and 'Yolk Volume'. Age was used as the only explanatory variable for 'DNA Damage' since the damage was only investigated at high UVR treatment conditions. All models were validated using MCMC diagnostic plots from the package 'bayesplot' (Gabry et al., 2022), and residuals were visually inspected using the 'DHARMa' package (Hartig and Hartig, 2017). *Post hoc* comparison tests were used to evaluate differences between UVR treatments using the emmeans function in the 'emmeans' package (Lenth et al., 2022). Predictions from the best-fitting model were compiled using the package 'tidybayes' (Kay and Mastny, 2022). The data and model outputs were visualised using the packages 'ggplot2' (Wickham et al., 2021), 'ggdist' (Kay and Wiernik, 2023), and 'cowplot' (Wilke, 2020).

## Acknowledgements

We acknowledge the Traditional Owners and their custodianship of the lands on which this study was conceived and experiments were performed. We thank the UQBR aquarium staff for helping with animal husbandry and setting up the experimental tanks. We also acknowledge The Company of Biologists and the Goodman Foundation for providing financial support to A.T.D.

## Competing interests

C.E.F. is Editor-in-Chief of Journal of Experimental Biology, published by The Company of Biologists. C.E.F. was not involved in determining the funding provided to A.T.D. by The Company of Biologists. We declare no other competing interests.

## Author contributions

Conceptualization: A.T.D., R.L.C., F.C., C.E.F.; Data curation: A.T.D.; Formal analysis: A.T.D.; Funding acquisition: A.T.D., F.C., C.E.F.; Investigation: A.T.D., C.H., R.L.C., F.C., C.E.F.; Methodology: A.T.D., C.H., R.L.C., F.C., C.E.F.; Project administration: F.C., C.E.F.; Resources: R.L.C., F.C., C.E.F.; Supervision: R.L.C., F.C., C.E.F.; Writing – original draft: A.T.D.; Writing – review & editing: A.T.D., C.H., R.L.C., F.C., C.E.F.

## Funding

This work was supported by the Australian Research Council (ARC) Discovery grant (DP190102152) to C.E.F. and an ARC Discovery Early Career Research Fellowship (ARC DE200100620) to F.C. A.T.D. was supported by The Company of Biologists Travel Fellowship, a Goodman Research Foundation Grant, and The University of Queensland. F.C. was supported by an ARC Future Fellowship (FT240100725). Open Access funding provided by University of Queensland. Deposited in PMC for immediate release.

## Data and resource availability

Data (i.e. datasets and metadata for statistical analyses) used for this research are openly available via the Open Science Framework (DOI: 10.17605/OSF.IO/NBUK3).

## First Person

This article has an associated First Person interview with the first author of the paper.

## Peer review history

The peer review history is available online at https://journals.biologists.com/bio/lookup/doi/10.1242/bio.062107.reviewer-comments.pdf

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
