## [Peer Review File · Biology Open]

Ultraviolet B Radiation Impairs Coral Reef Fish Development

Coen Hird, Rebecca L. Cramp, Fabio Cortesi, Craig E. Franklin and Adam Downie

DOI: 10.1242/bio.062107

Editor: Lewis Halsey

Review timeline

Original submission: 10 June 2025

Editorial decision: 17 June 2025

First revision received: 30 July 2025

Accepted: 30 July 2025

Original submission

First decision letter

MS ID#: bio.062107

MS Title: Ultraviolet B Radiation Impairs Coral Reef Fish Development

Authors: Coen Hird, Rebecca L. Cramp, Fabio Cortesi, Craig E. Franklin and Adam Downie

I have now reached a decision on the above manuscript.

The reviewer reports are shown at the bottom of this email or can be accessed, together with a copy of this decision letter, by going to:

As you will see, the reviewers raised a number of substantial criticisms that prevent me from accepting the paper at this stage.

They suggest, however, that a revised version might prove acceptable, if you can address their concerns. If you think that you can deal satisfactorily with the criticisms on revision, I would be pleased to see a revised manuscript. We would then return it to the reviewers.

At this stage, we also ask you to ensure your manuscript complies with our formatting guidelines. Provided you are able to fully address the referees' comments, we are positive about publication of your paper (we accept over 95% of revision submissions) and therefore hope you won't mind any extra work involved in reformatting your manuscript at this point.

Please ensure that you clearly highlight all changes made in the revised manuscript. Please avoid using 'Tracked changes' in Word files as these are lost in PDF conversion.

I should be grateful if you would also provide a point-by-point response detailing how you have dealt with the points raised by the reviewers in the 'Response to Reviewers' box. Please attend to all of the reviewers' comments. If you do not agree with any of their criticisms or suggestions please explain clearly why this is so.

Reviewer 1

Comments for the author

The authors have exposed clownfish embryos to two levels of UVB radiation corresponding to the levels at 2 and 6 meters water depth, mimicking what could happen in a future coral reef with less structures shielding against (i.e. habitat loss) UV radiation. The authors should be commended for identifying this largely over-looked threat for marine organisms, particularly for those living in low-latitude clear waters, like tropical coral reefs. The results are quite striking since all UV exposed embryos died before hatching, with the high dose group also showing DNA damage.

I have only some minor comments for the authors:

Line 208. Here the term "dead eggs" is used while in other places "dead embryos" is used (similarly on line 213). Please be consistent in the use of the words eggs and embryos.

Also, I find no mentioning of the fertilization success. Were all eggs fertilized?

Please describe how an embryo was identified as dead.

Line 279. Even if it is stated in the figure legend that no DNA damage was detected in controls and the low UVR group, I think that it should also be stated in the results. Moreover, make it clear in the results that the statement on line 279 refers to the high UVR group.

Line 290. It is stated "contrary to our hypotheses, we found that embryo exposure to high and low UVR resulted in increased yolk sac volume at 2 dpf and 5 dpf". Why would this not be expected if the embryos are dying and therefore not able to utilize the yolk?

When discussing the lack of detectable DNA damage in the low UVR group, although these embryos still died, it may be worth pressing at the possibility that other cellular components can be damaged by UVR, as the authors do point out in the Introduction: "UVR is both cytotoxic (increases production of oxygen radicals, which causes oxidative damage to cells and metabolites) and genotoxic (directly damages DNA)".

Lines 391 and 394. Please use "ATD" in both instances.

Fig. 2. Please explain the colours of the lines (i.e. which group is which).

Reviewer 2

Comments for the author

General comments:

This manuscript examines the effect of two levels of ecologically relevant UV radiation levels on clownfish embryos. The experimental work is interesting, useful and sound. However, the manuscript writing itself feels rushed and most of my comments are simple editorial fixes. I have only two major comments, and many suggestions to help improve the clarity of the methods.

First, the manuscript somewhat suffers from the experimental results being "too good". The main result - 0% mortality in control and 100% mortality in treatments - is the strongest effect possible (and the strongest effect I have ever seen). I think the discussion needs to address this, for example perhaps the experimental protocol didn't represent real conditions. This may include: a) was the pot fragment facing up directly to UV light, which wouldn't happen in the wild and embryos have basically zero capacity to tolerate exposure to any UV light? Or b) fault in the equipment used to test UV exposure? I'm not suggesting that the results are invalid, and I understand why the experiment was designed this way, but the discussion should acknowledge this stark reality and provide some more context on the fact that 100% mortality at even the lower UV level was observed. Clearly the study species would have zero reproductive success in these conditions, so how likely would it be for wild clownfish to experience no shelter at all even with degradation of coral reefs?

Second, it wasn't clear to me why high UV exposure had less of an effect than low UV exposure on yolk volume. Perhaps it is inconsequential given 100% mortality at all exposures, but again I think the discussion could be clarified on this point.

Overall, with some tidying up of the manuscript presentation this will be a useful contribution to the literature.

Specific comments:

- * Ln 10: somewhere in the abstract it would be useful to say how long the exposure to UVR was. Does 'exposed' mean constant exposure?
- * Ln 11-12: missing a closing parenthesis.
- * Ln 11: UVBR isn't defined.
- * Ln 16: I think 'UVRB' is a mistake but it may be another term not defined.
- * Ln 21: very minor formatting point but no capital needed for "Damage".
- * Ln 39: maybe "invertebrates" is more descriptive than "slugs and crabs"? It's a minor point but it's not clear why these two examples are singled out.
- * Ln 45: remove semi colon.
- * Ln 51: please check citation format/details.
- * Ln 67: remove comma.
- * Ln 75-78: are all the preceding references in temperate fish only? This point feels a little contrived, especially without any references. Are there really no studies on this in tropical fish? I assume some of the many Bellwood references already cited contain some information on this point.
- * Ln 99: why is the parental care aspect an important determining factor for study species choice?
- * Ln 127: Why was the pot fragmented? Why not just remove the whole pot and reduce risk or damage to the eggs? Is it because the shard was turned upside down for full UV exposure? Or were the eggs still on the underside of the shard and somewhat protected? Please clarify this.
- * Ln 146-147: So the control eggs were in complete darkness? Could this be a confounding factor?
- * Ln 158: but was there also non-UV lighting provided? Or were the eggs in darkness except for this 2-hour window of UV exposure?
- * Ln 160: please clarify if each clutch of eggs was separated into the 3 groups, or each clutch was used for a single treatment? It may help to state how many different clutches of eggs were used.
- * Ln 163: were the eggs exposed to air in this process?
- * Ln 166: reference to figure 2 is probably not needed in methods - figure 1 hasn't been referenced yet.
- * Ln 168: is "n = 9 per trait" correct? Should this be "per treatment"? or per clutch per treatment?
- * Ln 169-175: this section is not clear to me, and I think should be a new paragraph. Was this done to all eggs? Some eggs? And why? I think it needs a signpost sentence at the start to show the reader this is to assess DNA damage. After reading the later section in the methods subtitled 'DNA Damage', I think this text is better placed there, so the reader can easily follow the protocol for each metric.
- * Ln 174: "light-induced DNA photo repair" should have a reference.
- * Ln 187-188: this sentence belongs in the introduction.
- * Ln 189: Is this section about developmental progression different from the yolk sac volume point? I think this should be a new paragraph.
- * Ln 212: The denominator of this equation is a little confusing to me. Is 'a = 0' needed? Could this term just be C?
- * Ln 235: Should this be Figure 1B?
- * Ln 237: HPDI hasn't been defined.
- * Ln 236-238: This is slightly ambiguous - better to say directly which one is lower (i.e. control was lower than high UVR), especially because the dose-response relationship is unclear.
- * Ln 245: start a new paragraph for the new results section on embryo mass.
- * Ln 245-246: this statement looks incorrect - there seems to be a clear difference in embryo mass with age?
- * Ln 255: Why is panel C not showing controls and low UV treatment? They are mentioned in the text but not discussed. Also it looks like damage was higher at 2 dpf but the figure description

says damage only occurred at 5dpf (and the text says this is non-significant, which should be mentioned in the figure caption).

* Ln 263: I find the line colour for Control difficult to distinguish against the white background. Also the caption needs to explain what the different lines within treatment groups are (I assume clutches?).

* Ln 287-289: I think these opening statements are too weak given the results. I realise scientific writing should not exaggerate, but there was literally 100% mortality. I think that "survival was significantly reduced" undersells the work - survival wasn't just 'reduced', it was not observed.

* Ln 296: If this is consistent with previous work then why did it go against your hypothesis?

* Ln 299: delete "very".

* Ln 301: what is the genus? What type of organism is this? The reader hasn't been told.

* Ln 308 - 310: is this talking about your results now?

* Ln 321: Again the self-citation format is inconsistent or missing?

* Paragraph ending Ln 323: it's not immediately clear to me why the high UV had less of an effect on yolk diameter than the low UV treatment? I.e. why was there no dose-response relationship here?

* Ln 359: ROS not defined. Don't bother with an acronym that's only used once - just spell out the term.

* Ln 370-372: I'm not sure if there's literature available on this, but is it possible that anemones could increase under changing coral reef conditions? I.e. it's seen in Caribbean reefs that hard corals have experienced widespread decline, but soft coral and algae have proliferated. This is clearly detrimental for reef building, but could anemones fare better with reduced hard coral cover?

* Ln 372-374: this point on marine spatial planning feels contrived. I'm not sure how this could be incorporated based on this text - please either dedicate more text to explaining it or just delete this sentence.

* Ln 374-376: Again this feels contrived. What would the benefit of measuring UV be in this case? What's the management outcome? Are you proposing UV blocking strategies similar to what is being trialled to reduce coral bleaching?

* Ln 383-385: Again, this seems contrived. I think this knowledge is important and interesting in its own right - if the text is going to propose management or future research outcomes arising from it, I suggest dedicating a paragraph to each point, or just remove these throwaway sentences.

* Discussion: I think there is a missing section on the 'realism' aspect - these are extremely stark results, to the point that I question if clownfish would ever lay their eggs facing 'up' to UVR given they clearly have no evolutionary adaptation for this. Therefore the premise that loss of habitat means there will be no refuge for egg-laying ensures local extinction of this species.

Reviewer's Responses to Questions

Experimental quality

Does each figure have the proper controls?

If 'No', please indicate reasons in Comments for Author box below.

Reviewer #1:

- Yes

Reviewer #2:

- No

Were the data analyzed using appropriate statistical tests?

If 'No', please indicate reasons in Comments for Author box below.

Reviewer #1:

- Yes

Reviewer #2:

- Yes

Reproducibility

Were experiments performed using adequate number of biological replicates?
If 'No', please indicate reasons in Comments for Author box below.

Reviewer #1:

- Yes

Reviewer #2:

- No

Does the methods section provide sufficient detail to permit reproducibility?
If 'No', please indicate reasons in Comments for Author box below.

Reviewer #1:

- Yes

Reviewer #2:

- Yes

Completeness

Are the manuscript's conclusions supported by the data?
If 'No', please indicate reasons in Comments for Author box below.

Reviewer #1:

- Yes

Reviewer #2:

- No

Scholarship

Do the authors cite and discuss the merits of data that would argue for and against their conclusion?
If 'No', please indicate reasons in Comments for Author box below.

Reviewer #1:

- Yes

Reviewer #2:

- No

Does the manuscript title & abstract accurately reflect the contents of the manuscript, without hyperbole?

If 'No', please indicate reasons in Comments for Author box below.

Reviewer #1:

- Yes

Reviewer #2:

- No

First revision

Author response to reviewers' comments

*Note: reference to line changes are per the 'clean' version of the revised manuscript

Reviewer 1

The authors have exposed clownfish embryos to two levels of UVB radiation corresponding to the levels at 2 and 6 meters water depth, mimicking what could happen in a future coral reef with less structures shielding against (i.e. habitat loss) UV radiation. The authors should be commended for identifying this largely over-looked threat for marine organisms, particularly for those living in low-latitude clear waters, like tropical coral reefs. The results are quite striking since all UV exposed embryos died before hatching, with the high dose group also showing DNA damage.

We thank the reviewer for their support and constructive criticism of our manuscript.

I have only some minor comments for the authors:

Line 208. Here the term "dead eggs" is used while in other places "dead embryos" is used (similarly on line 213). Please be consistent in the use of the words eggs and embryos.

Thank you for this comment. For consistency's sake, we use 'dead embryos' throughout.

Also, I find no mentioning of the fertilization success. Were all eggs fertilized?

Successfully fertilized eggs have a light pink colour with a bright white vegetal pole/cap to them. The male swiftly removes all unfertilized eggs, which appear beige in colour, during mating. When the pots were assessed for eggs, the egg mass was generally completely pink. The following sentence was added:

Ln 134-138: "Newly fertilized eggs are a light pink with a bright white vegetal pole, while unfertilized eggs appear beige in colour (Salis et al 2021). The male clownfish recognises unfertilised eggs and removes them swiftly during mating. Egg masses used in the experiments were typically all pink with white caps, indicating that all eggs were fertilized."

Please describe how an embryo was identified as dead.

Good point, and we have clarified with the following sentence:

Ln 192-195: "Embryos are typically a light grey colour with the eyes having a silvery sheen (Salis et al 2021). Dead embryos were therefore easily identifiable because the body was white and the eyes were no longer shiny".

Line 279. Even if it is stated in the figure legend that no DNA damage was detected in controls and the low UVR group, I think that it should also be stated in the results. Moreover, make it clear in the results that the statement on line 279 refers to the high UVR group.

We agree that there should have been a mention of no DNA damage measured in low UVR exposed embryos. The following sentence was added:

Ln 314: “Interestingly, there was no measurable DNA damage in low UVR-exposed embryos.”

Line 290. It is stated “contrary to our hypotheses, we found that embryo exposure to high and low UVR resulted in increased yolk sac volume at 2 dpf and 5 dpf”. Why would this not be expected if the embryos are dying and therefore not able to utilize the yolk?

This is a good point, however, the embryos aren’t necessarily all dead at these ages. They would still be alive and requiring energy from the yolk to develop and respond to the UVR stress. Under these circumstances, we would expect the yolk to decrease in volume, not remain the same or increase.

When discussing the lack of detectable DNA damage in the low UVR group, although these embryos still died, it may be worth pressing at the possibility that other cellular components can be damaged by UVR, as the authors do point out in the Introduction: “UVR is both cytotoxic (increases production of oxygen radicals, which causes oxidative damage to cells and metabolites) and genotoxic (directly damages DNA)”.

We have modified the following sentence to reiterate the genotoxic and cytotoxic nature of UVR:

Ln 435-440: “Given that UVR is both genotoxic and cytotoxic, other cellular and molecular processes may have been impaired, which contributed to the elevated mortality rate in exposed embryos. For example, UVR may have denatured the hatching enzymes that help dissolve the egg casing, or oxidative stress (e.g., ROS build-up) causes additional cellular stress, diverting energy from development and hatching.”

Lines 391 and 394. Please use “ATD” in both instances.

Changes made.

Fig. 2. Please explain the colours of the lines (i.e. which group is which).

The figure caption has been modified to explicitly describe which lines correlate with which experimental group:

“Figure 2. The impact of ultraviolet radiation (UVB 280-315 nm) on embryo cumulative survival (%) across developmental age (days post fertilization; dpf) of clownfish (*Amphiprion ocellaris*). Embryos were exposed to either control (no UVR; pink line) or UVR levels representing the within-depth range of *A. ocellaris*: 6m (low UVR treatment; $80 \mu\text{W m}^{-2}$; green line) or 2 m (high UVR treatment; $280 \mu\text{W m}^{-2}$; blue line). UVR exposure occurred for 2 h daily from when eggs were laid (0 dpf) until hatching at 8-9 dpf. The experiment was replicated three times per treatment using two different parent pairs, represented by three lines per treatment.”

Reviewer 2

This manuscript examines the effect of two levels of ecologically relevant UV radiation levels on clownfish embryos. The experimental work is interesting, useful and sound. However, the manuscript writing itself feels rushed and most of my comments are simple editorial fixes. I have only two major comments, and many suggestions to help improve the clarity of the methods.

We thank the reviewer for their support and constructive criticism of our manuscript.

First, the manuscript somewhat suffers from the experimental results being “too good”. The main result - 0% mortality in control and 100% mortality in treatments - is the strongest effect possible (and the strongest effect I have ever seen). I think the discussion needs to address this, for example perhaps the experimental protocol didn’t represent real conditions. This may include: a) was the pot fragment facing up directly to UV light, which wouldn’t happen in the wild and embryos have basically zero capacity to tolerate exposure to any UV light? Or b) fault in the equipment used to test UV exposure? I’m not suggesting that the results are invalid, and I

understand why the experiment was designed this way, but the discussion should acknowledge this stark reality and provide some more context on the fact that 100% mortality at even the lower UV level was observed. Clearly the study species would have zero reproductive success in these conditions, so how likely would it be for wild clownfish to experience no shelter at all even with degradation of coral reefs?

This is a valid concern, and we understand that the stark contrast between 100% survival under control and 0% survival under UVR should be addressed. In terms of the 100% survival rate, when provided adequate aeration, clownfish embryos that survive to hatching age typically have a 100% hatch rate (ATD personal observation). In other experiments involving impacts of UVR, > 90-100% survival of control animals is observed (see: Lundsgaard et al 2020 <https://doi.org/10.1093/conphys/coaa002>; Lundsgaard et al 2024; <https://doi.org/10.1002/jez.2882>). The following sentence has been added to address the 100% survival:

Ln 361-368: “In support of our hypothesis, UVR had significant impacts on *A. ocellaris* embryo survival, which is also commonly found in embryos of other fish species exposed to UVR (Holmquist et al., 2014; Kouwenberg et al., 1999; Lesser et al., 2001; Mahmoud et al., 2009). Under control scenarios, embryos experienced no mortality. When provided adequate aeration, it is common for all clownfish embryos to survive and hatch (ATD personal observation). Similar experiments exposing amphibians to UVR have also found that under control conditions, 90 to 100% of individuals survived (Lundsgaard et al., 2020; Lundsgaard et al., 2024).”

Regarding the second point, the equipment has been used, calibrated and tested for numerous experiments, so faulty equipment is highly unlikely to have caused the strong outcomes we observed. The experiments followed a standard procedure of exposing fish embryos directly to UVR, following the protocols from Araujo et al (2021; <https://doi.org/10.1016/j.scitotenv.2020.142899>), Vasquez et al (2016; <https://doi.org/10.1071/MF14038>), and Mekkawy et al (2010; <https://doi.org/10.1007/s10695-009-9334-6>). This experimental approach is designed to understand the mechanism behind embryonic responses to UVR (typically by measuring enzymes and other molecular indicators), and is not necessarily ecologically relevant. However, as indicated in the introduction, embryos of these fishes could be exposed to prolonged UVR exposure for various reasons in nature, so the question remains valid. Studies exploring the effects of UVR exposure on embryos from different fish species have found highly variable responses to UVR in terms of survival, which we now detail:

Ln 368-392: “In stark contrast, under both high and low UVR, clownfish embryos experienced 100% mortality. Typically, mortality rates of fish embryos exposed to UVR are highly variable and are also dependent on UVR dosage and duration of experiments. For example, embryos of Woundfin (*Plagopterus argentissimus*) and Yellow perch (*Perca flavescens*) experienced 100 and 98% mortality, respectively, when exposed to UVR (Holmquist et al., 2014). On the contrary, mortality of anchovy embryos (*Engraulis ringens*) ranged from 24% to 53% (Vasquez et al 2016), depending on embryonic stage and UVR dosage (UVA only versus UVB and UVA). Atlantic cod (*Gadus morhua*) embryos experienced 80% survival when exposed to UVR intensity of 6.38 Wm² (total dosage of 472 kJ) for 30 min, and 27% survival when exposed to UVR intensity of 4.04 Wm² (total dosage of 116.4 kJ) for 2.5 hours (Kouwenberg et al 1999). Tropical and subtropical species also have been found to have high embryonic mortalities when exposed to UVR, such as up to 75% for zebrafish (*Danio rerio*; Dong et al 2007) and up to 90% for the common sardine (*Strangomera bentincki*; Vasquez et al 2016). Given the variable response of fish embryo survival to UVR, species-specific responses to UVR at the cellular level are likely at play (e.g., photo-protection, antioxidant responses, use of alternate metabolic pathways to supplement energy in response to stress). Interestingly, both low and high UVR treatments resulted in 100% mortality in clownfish embryos, with most mortalities occurring before hatching. Similar to what is causing the differences in yolk volume between high and low UVR treatments, it is possible that the higher UVR treatment elicited a more immediate biochemical response, which may have increased energy use early on, resulting in less energy for survival as the embryos approached hatching. In contrast, the low UVR treatment is likely to have caused a more gradual accumulation of damage that surpassed a tolerance threshold at hatching. It is also likely that hatching enzymes in both treatments were damaged or depleted, which would cause a systematic loss of the clutch, i.e., total mortality.”

The vulnerability of embryos to both high and low UVR suggests anemonefish embryos may not have the biochemical defenses to protect against prolonged exposure to UVR. Through these experiments we aren't trying to demonstrate reality - the reality for these animals is never experiencing UVR - as evidenced by their very high susceptibility to even very low UVR levels. Instead, we are trying to understand what would happen if they did get exposed to ecologically realistic UVR levels.

The direct impact of reef degradation on clownfish reproduction and future generation success has only recently been investigated and there is clearly a major knowledge gap that needs to be filled. Recent evidence suggests that anemonefish populations are highly susceptible to short-term and lesser-scale changes in habitat structure (e.g., decreases in anemones associated with environmental stressors) (Hobbs et al 2013; Salles et al 2020). Additionally, fecundity is decreased when bleached anemones are present (Belade et al, 2017), and adults spend less time in bleached anemones (Cortese et al, 2021), which may have carry-over effects on the quality of embryos produced/hatching rate and larval survival. The impacts of UVR as an additional stressor are likely to compound the effects of a degraded habitat. The degree to which suitable habitat is available for reproduction also depends on location, as bleaching along reefs is variable, at least along the Great Barrier Reef (Hughes et al, 2017). Therefore, the extent to which available shelter/habitat increases competition for space, resources, and reproduction is of interest. We have added the following text:

Ln 411-429: “Our results contribute to the growing body of literature that suggests that clownfishes are highly sensitive to environmental change across life history stages, and reproductive output may be impacted by environmental stressors. Indeed, scientists have only just started to investigate the direct impact of coral cover, anemone availability, and habitat quality on clownfish reproduction and carry-on effects for future generations (Salles et al 2020). Macro-ecological studies have found that clownfish populations are susceptible to acute, small-scale changes in habitat structure and quality, such as decreases in anemone abundance associated with ocean warming and irradiance (Hobbs et al, 2013; Salles et al 2020). This loss of shelter would presumably expose early life stages to adverse environmental conditions. Bleaching is a significant threat to coral reefs and contributes significantly to increased habitat degradation of the world’s coral reefs (Hughes et al 2017). Recent evidence suggests that under anemone bleaching conditions, fecundity is decreased by up to 73% (Beldade et al 2017). Additionally, adult clownfishes spend less time in bleached anemones (Cortese et al 2021), which would have ramifications on the parental investment in embryo development. A pervasive stressor such as UVR is likely to compound the effects of bleaching and habitat degradation, especially if eggs are directly exposed to the radiation. Future work evaluating the reproductive output of marine life under various degrees of habitat degradation and environmental stressors (including UVR) will provide valuable insights into the significance of shelter for preserving populations under stress.”

Second, it wasn't clear to me why high UV exposure had less of an effect than low UV exposure on yolk volume. Perhaps it is inconsequential given 100% mortality at all exposures, but again I think the discussion could be clarified on this point.

This is an excellent point, and we now discuss this aspect in more detail in the revised version. Ln 332-342: “Interestingly, low UVR had a greater effect on yolk volume than high UVR (low and high UVR-treated embryos had 61% and 38% greater yolk volume than controls, respectively). This may be attributed to the initial biochemical response of embryos to high versus low UVR. Higher UVR dosages may have elicited a greater biochemical response and therefore invested more energy in antioxidant production, DNA damage and photorepair. In contrast, low UVR exposure may elicit less energy-intensive responses; however, these responses accumulate over the entire embryonic cycle, resulting in an overall high mortality rate.”

Overall, with some tidying up of the manuscript presentation this will be a useful contribution to the literature.

We appreciate the concerns raised by the reviewer and feel that by addressing them, we have significantly improved our manuscript.

Specific comments:

* Ln 10: somewhere in the abstract it would be useful to say how long the exposure to UVR was. Does 'exposed' mean constant exposure?

The following sentence has been modified:

Ln 9-11: “Here, using laboratory experiments, we exposed embryos of the clownfish *Amphiprion ocellaris* for two hours daily to two UVR levels reflective of their depth at settlement...”

* Ln 11-12: missing a closing parenthesis.

Closed parenthesis added.

*** Ln 11: UVBR isn't defined.**

Yes, this is correct. UVBR is an acronym for 'Ultraviolet B Radiation' which was the wavelength used for the experiment. For simplicity, we kept it at UVR.

*** Ln 16: I think 'UVRB' is a mistake but it may be another term not defined.**

Correct. We have made the changes from UVBR to UVR.

*** Ln 21: very minor formatting point but no capital needed for "Damage".**

Corrected.

*** Ln 39: maybe "invertebrates" is more descriptive than "slugs and crabs"? It's a minor point but it's not clear why these two examples are singled out.**

We agree, and the following sentence has been modified accordingly:

Ln 38-42: "Since hard corals provide structure and refuge for many reef-associated animals, including invertebrates and fishes, their loss poses significant challenges to survival."

*** Ln 45: remove semi colon.**

Semi-colon removed.

*** Ln 51: please check citation format/details.**

We have checked the formatting throughout. For some reason Endnote has added in the second author for some of my papers if the papers are from the same date (e.g., Downie, Wu et al 2023, Downie, Lefevre et al 2023).

*** Ln 67: remove comma.**

Comma removed.

*** Ln 75-78: are all the preceding references in temperate fish only? This point feels a little contrived, especially without any references. Are there really no studies on this in tropical fish? I assume some of the many Bellwood references already cited contain some information on this point.**

The sentence has been removed:

"Therefore, prolonged UVR exposure during the embryonic phase may significantly impact population dynamics by creating a larval recruitment bottleneck, at least in temperate species."

The Bellwood references are there to highlight that sheltering behaviour is common in reef fishes during peak sunlight. To make the references clearer, we have placed the references beside the biological traits they are citing. The sentence now reads:

Ln 56-60: "Tropical fishes have evolved many strategies to cope with high UVR intensities, such as biological sunscreens in the skin and eyes (mycosporine-like amino acids, MAAs; Mason et al., 1998), antioxidant defences (Banaszak & Lesser, 2009), photo-repair mechanisms (Dethlefsen et al., 2001; Dong et al., 2007), and behaviourally avoiding UVR by seeking refuge under tabular corals (Kerry & Bellwood, 2012, 2015a, 2015b, 2017)."

*** Ln 99: why is the parental care aspect an important determining factor for study species choice?**

We agree that the parental care aspect per se isn't important for making our point. The point is that fish with parental care tend to invest more energy into fewer, well-developed eggs than pelagic spawning fishes. The following sentence has been altered:

Ln 85-94: "While many tropical species broadcast thousands or millions of pelagic eggs in a spawning event, some tropical species, such as clownfishes (Amphiprioninae, Pomacentridae), lay fewer (hundreds), well-developed eggs among anemones and coral to protect them from environmental conditions, including UVR (Pacaro et al., 2023). While pelagic eggs are exposed to UVR, some species have been found to biochemically initiate negative buoyancy when UVR exposure crosses a threshold (Dethlefsen et al 2002, Pasparakis et al 2013). In contrast, benthic eggs are typically adhered to the substrate and are unable to move from stressors, relying solely on structures like coral and anemones for shelter."

We also removed the emphasis on parental care from formerly Ln 99 (now Ln 105-106).

*** Ln 127: Why was the pot fragmented? Why not just remove the whole pot and reduce risk or damage to the eggs? Is it because the shard was turned upside down for full UV exposure? Or were the eggs still on the underside of the shard and somewhat protected? Please clarify this.**

It had to be fragmented because the entire pot did not fit in the tanks designated for UVR exposure. The following sentence was added:

Ln 140-143: “Effort was made that fragmenting the pot did not create a fissure through the clutch. Instead, the pot was broken around the clutch so that it was a small enough size to fit into the experimental tank, as the experimental tank was too narrow to accommodate the entire pot.”

*** Ln 146-147: So the control eggs were in complete darkness? Could this be a confounding factor?**

No, the control eggs were not in darkness and were exposed to the same natural photoperiod as the experimental eggs provided by ceiling fluorescent lights. Some of the surrounding tanks in our lab were outfitted with overhead lights that generated low levels of UV-A and UV-B wavelengths. Therefore, to reduce the potential influence of these lights, a sheet of plexiglass was placed on top of the experimental tank when control eggs were in the experimental tank. This plexiglass allowed ambient lighting from the ceiling (fluorescent bulbs with no UVR emitted) over the natural photoperiod. The sides of the tanks were also covered with black plastic to block adjacent UVR lighting. However, the top of the tank was open and the sheets of Plexiglas or thin plastic completely allowed ambient light (i.e., no UVR) from the ceiling lights to enter the tank and not disrupt the natural photoperiod. We understand this wording was confusing, so the following sentence has been modified with these details:

Ln 165-173: “Control embryos were placed in the experimental tank without UV lamps turned on (i.e., UVB: $0 \mu\text{W m}^{-2}$) during the designated exposure times (see below). Because low levels of UV lighting were always present in the aquarium room (rearing tanks in the same room had UVR lights installed and emitted low levels of UV-A and UV-B), control tanks were covered with a sheet of UV-absorbing plexiglass placed on top of the tank. Additionally, the sides of the tank were covered with black plastic to prevent any potential UVR from entering from the sides of the tank. Hence, control embryos were exposed to the same photoperiod as the two treatments, experiencing a UV-deprived light habitat in the human visible spectrum produced from the fluorescent ceiling lights.”

*** Ln 158: but was there also non-UV lighting provided? Or were the eggs in darkness except for this 2-hour window of UV exposure?**

See the reply to the previous comment.

*** Ln 160: please clarify if each clutch of eggs was separated into the 3 groups, or each clutch was used for a single treatment? It may help to state how many different clutches of eggs were used.**

This is a good point. Each individual clutch was designated as either control, low UVR or high UVR. We did this for a few reasons. Firstly, it would have been too difficult to fragment the clutch into three equal pieces for each treatment. Fragmenting the pot generally resulted in a single large piece with all the eggs on it. Breaking the pot further to split the clutch would have likely caused embryo mortality, and as the terracotta pots do not have set lines, the fragments would have been uneven. Secondly, a clutch consisted of a few hundred eggs. If we divided the clutch into thirds then the remaining eggs per treatment fragment would have been too low of a sample size to account for mortality, and collected metrics (morphology and DNA damage). The following sentences were added for clarification.

Ln 184-187: “Each experimental group was replicated three times. A single clutch was used at a time for each experimental treatment replicate, spawned from one of two breeding pairs. Clutches were not divided as there was no guarantee that clutches would fragment evenly and directly fragmenting through the clutch may have caused mortalities.”

*** Ln 163: were the eggs exposed to air in this process?**

No, the eggs were not exposed to air during this process. Care was made that the eggs were transferred from the experimental tank into a small dish filled with water from the experimental tank, which was then placed on a bench and briefly evaluated for dead embryos. This has been clarified:

Ln 188-192: “Every morning (approximately 07:00), the terracotta fragments with developing embryos were gently removed from the experimental tank using a small dish containing water from the experimental tank and placed on a benchtop to evaluate mortalities under a benchtop dissection microscope. Embryos were not exposed to air during this process, as transfer of the fragment with embryos into the small dish occurred underwater.”

*** Ln 166: reference to figure 2 is probably not needed in methods - figure 1 hasn't been referenced yet.**

This is true. We added '(see Results for mortalities)' instead.

*** Ln 168: is "n = 9 per trait" correct? Should this be "per treatment"? or per clutch per treatment?**

Yes, this is n=9 per developmental age (2 and 5 dpf) for each treatment and replicate. This has been clarified:

Ln 198-199: "At 2 and 5 dpf, individual eggs were collected to determine embryo mass, yolk volume, and DNA damage (n = 9 per trait per treatment replicate)."

*** Ln 169-175: this section is not clear to me, and I think should be a new paragraph. Was this done to all eggs? Some eggs? And why? I think it needs a signpost sentence at the start to show the reader this is to assess DNA damage. After reading the later section in the methods subtitled 'DNA Damage', I think this text is better placed there, so the reader can easily follow the protocol for each metric.**

This text justifies why we selected the ages post-fertilization. For clarity, the following sentence has been modified:

Ln 204-205: "Embryos were carefully removed from the egg casings using forceps to be evaluated for DNA damage and morphology."

*** Ln 174: "light-induced DNA photo repair" should have a reference.**

The following references have been provided:

Ln 205-208: "Embryos for DNA damage assays were immediately placed in dark tubes to prevent light-induced DNA photo-repair (Kelner, 1949; Rupert et al., 1958; Sancar, 2008) and stored at -80 °C and embryos measured for mass and yolk volume were placed in phosphate-buffered seawater and stored in the fridge at 4 °C."

*** Ln 187-188: this sentence belongs in the introduction.**

We think the information is relevant for the methods because it justifies the selection of ages post-fertilization. However, we did add this information in the introduction as well:

Ln 51-55: "While UVR typically targets cellular processes and molecules, the impacts at the cellular level cascade to impact the whole animal, resulting in several sub-lethal and lethal consequences on animal fitness, such as delayed developmental stages (Lundsgaard et al., 2021; Lundsgaard et al., 2024), decreased locomotion (Kazerouni et al 2015), and mortality (Holmquist et al 2014)."

*** Ln 189: Is this section about developmental progression different from the yolk sac volume point? I think this should be a new paragraph.**

Similar to our point above, the reference here is to justify why we selected the ages that we did.

*** Ln 212: The denominator of this equation is a little confusing to me. Is 'a = 0' needed? Could this term just be C?**

The zero indicates the day the eggs were fertilized (i.e., 0 dpf). There are several other spots in the equation where a (age) is at any given age. The distinction is needed to be able to solve the equation.

*** Ln 235: Should this be Figure 1B?**

Changes made.

*** Ln 237: HPDI hasn't been defined.**

HPDI has been defined as upper or lower Highest Posterior Density Interval.

*** Ln 236-238: This is slightly ambiguous - better to say directly which one is lower (i.e. control was lower than high UVR), especially because the dose-response relationship is unclear.**

The sentence has been changed for clarity:

Ln 268-271: "At 2 dpf, average embryo yolk sac volume was 29% lower under control conditions than under high UVR (HPDI_{lower} = -0.36, HPDI_{upper} = -0.034), and yolk sac volume was 48% lower under control conditions than low UVR treatments (HPDI_{lower} = -0.48, HPDI_{upper} = -0.15; Figure 2B)."

* Ln 245: start a new paragraph for the new results section on embryo mass.

We have added new paragraphs for each metric in the Results section.

* Ln 245-246: this statement looks incorrect - there seems to be a clear difference in embryo mass with age?

Statistically, these results are correct.

* Ln 255: Why is panel C not showing controls and low UV treatment? They are mentioned in the text but not discussed. Also it looks like damage was higher at 2 dpf but the figure description says damage only occurred at 5dpf (and the text says this is non-significant, which should be mentioned in the figure caption).

The control and low UVR embryos did not have any measurable evidence for DNA damage, so they were not graphed. The mention of 5 dpf in the figure caption was a typo and has been removed.

* Ln 263: I find the line colour for Control difficult to distinguish against the white background. Also the caption needs to explain what the different lines within treatment groups are (I assume clutches?).

The line colours have been changed, and the figure caption now better describes which lines correlate to which experimental group.

* Ln 287-289: I think these opening statements are too weak given the results. I realise scientific writing should not exaggerate, but there was literally 100% mortality. I think that "survival was significantly reduced" undersells the work - survival wasn't just 'reduced', it was not observed.

The opening statements have been changed as follows:

Ln 292-295: "Exposure to UVR had significant consequences for the embryonic stage of the False Percula Clownfish, causing complete mortality independent of exposure levels. This was unexpected, as we assumed UVR would cause damage but not the complete loss of a clutch, especially at lower exposure."

* Ln 296: If this is consistent with previous work then why did it go against your hypothesis?

For clarity, we changed the following sentence:

"UVR had an impact on the development of clownfish embryos, particularly yolk volume."

* Ln 299: delete "very".

Removed.

* Ln 301: what is the genus? What type of organism is this? The reader hasn't been told.

The common name (African Sharptooth Catfish) and genus (*Clarias gariepinus*) have been added.

* Ln 308 - 310: is this talking about your results now?

We have modified the sentence for clarity:

Ln 314-316: "In the current study, it is possible that UVR may have denatured proteins and enzymes that maintained osmotic balance in the yolk sac, causing inflation; however, the yolk contents would have to be measured to confirm."

* Ln 321: Again the self-citation format is inconsistent or missing?

These citations have been fixed.

* Paragraph ending Ln 323: it's not immediately clear to me why the high UV had less of an effect on yolk diameter than the low UV treatment? I.e. why was there no dose-response relationship here?

The results are a bit surprising and would require further biochemical or molecular measurements, which were outside the scope of this paper. We do offer some hypotheses as to what may have occurred:

Ln 330-337: "Interestingly, low UVR had a greater effect on yolk volume than high UVR (low and high UVR-treated embryos had 61% and 38% greater yolk volume than controls, respectively). This may be attributed to the initial biochemical response of embryos to high versus low UVR. Higher UVR dosages may have elicited a greater biochemical response and therefore invested more energy in antioxidant production, DNA damage and photorepair. In contrast, low UVR exposure may elicit

less energy-intensive responses; however, these responses accumulate over the entire embryonic cycle, resulting in an overall high mortality rate.”

*** Ln 359: ROS not defined. Don't bother with an acronym that's only used once - just spell out the term.**

Acronym removed.

*** Ln 370-372: I'm not sure if there's literature available on this, but is it possible that anemones could increase under changing coral reef conditions? I.e. it's seen in Caribbean reefs that hard corals have experienced widespread decline, but soft coral and algae have proliferated. This is clearly detrimental for reef building, but could anemones fare better with reduced hard coral cover?**

This is an interesting point. Corals and anemones do compete for space, but there is no evidence of large phase-shifts from coral to anemone-dominated reefs when reef cover decreases. The warm temperatures that bleach reefs would also bleach anemones. We added some text on the impacts that habitat change would have on the reproductive output of anemonefishes.

Ln :418-423: “Bleaching is a significant threat to coral reefs and contributes significantly to increased habitat degradation of the world’s coral reefs (Hughes et al 2017). Recent evidence suggests that under anemone bleaching conditions, fecundity is decreased by up to 73% (Beldade et al 2017). Additionally, adult clownfishes spend less time in bleached anemones (Cortese et al 2021), which would have ramifications on the parental investment in embryo development.”

*** Ln 372-374: this point on marine spatial planning feels contrived. I'm not sure how this could be incorporated based on this text - please either dedicate more text to explaining it or just delete this sentence.**

We agree and removed this from the text.

*** Ln 374-376: Again this feels contrived. What would the benefit of measuring UV be in this case? What's the management outcome? Are you proposing UV blocking strategies similar to what is being trialled to reduce coral bleaching?**

This sentence has been removed.

*** Ln 383-385: Again, this seems contrived. I think this knowledge is important and interesting in its own right - if the text is going to propose management or future research outcomes arising from it, I suggest dedicating a paragraph to each point, or just remove these throwaway sentences.**

We believe acknowledging the growing body of literature that shows UVR interacts with other stressors is important and worth mentioning.

*** Discussion: I think there is a missing section on the 'realism' aspect - these are extremely stark results, to the point that I question if clownfish would ever lay their eggs facing 'up' to UVR given they clearly have no evolutionary adaptation for this. Therefore the premise that loss of habitat means there will be no refuge for egg-laying ensures local extinction of this species.**

The vulnerability of embryos to both high and low UVR suggests anemonefish embryos may not have the biochemical defenses to protect against prolonged exposure to UVR. As mentioned above, through these experiments we aren't trying to demonstrate reality - the reality for these animals is never experiencing UVR - as evidenced by their very high susceptibility to even very low UVR levels. Instead, we are trying to understand what would happen if they did get exposed to ecologically realistic UVR levels.

Second decision letter

MS ID#: bio.062107R1

MS Title: Ultraviolet B Radiation Impairs Coral Reef Fish Development

Authors: Coen Hird, Rebecca L. Cramp, Fabio Cortesi, Craig E. Franklin and Adam Downie

I've had the chance this morning to read through your very thorough rebuttal, and the associated manuscript edits. Clearly the key issue was the 'extreme' effect size that you've reported and I'm pleased that you've been able to deal with this clearly and fully, along with all the other comments made by the two Reviewers. In turn, I am happy to tell you that your manuscript has been accepted for publication in Biology Open, pending our standard publication integrity checks. It was accepted on 30 July 2025.